# A systematic review on the detection of volatile organic compounds in exhaled breath in experimental animals in the context of gastrointestinal and hepatic diseases

Kim F.H. Hintzen[1,2,3]☯, Myrthe M.M. Eussen[1]☯, Céline Neutel[1], Nicole D. Bouvy[1,4], Frederik-Jan van Schooten[2,3], Carlijn R. Hooijmans[5], Tim Lubbers[1,4]*

1 Department of Surgery, Maastricht University Medical Centre, Maastricht, The Netherlands, 2 Department of Pharmacology and Toxicology, Maastricht University, Maastricht, The Netherlands, 3 NUTRIM School of Nutrition and Translational Research in Metabolism, Maastricht University, The Netherlands, 4 GROW School for Oncology and Developmental Biology, Maastricht University, Maastricht, The Netherlands, 5 Department of Anesthesiology, Pain and Palliative Care (Meta Research Team), Radboud University Medical Centre, Nijmegen, The Netherlands

☯ These authors contributed equally to this work.
* tim.lubbers@mumc.nl

## Abstract

### Background

Analysis of volatile organic compounds (VOCs) in exhaled breath has the potential to serve as an accurate diagnostic tool for gastro-intestinal diseases. Animal studies could be instrumental as a preclinical base and subsequent clinical translation to humans, as they are easier to standardize and better equipped to relate specific VOCs to metabolic and pathological processes. This review provides an overview of the study design, characteristics and methodological quality of previously published animal studies on analysis of exhaled breath in gastrointestinal and hepatic diseases. Guidelines are provided for standardization in study design and breath collection methods to improve comparability, avoid duplication of research and reduce discomfort of animals in future studies.

### Methods

PubMed and Embase database were searched for animal studies using exhaled breath analysis to detect gastro-intestinal diseases. Risk of bias was assessed using the SYRCLE's risk of bias tool for animal studies. Information on study design, standardization methods, animal models, breath collection methods and identified VOCs were extracted from the included studies.

### Results

10 studies were included (acute liver failure n = 1, non-alcoholic steatohepatitis n = 1, hepatic ischemia n = 2, mesenteric ischemia n = 2, sepsis and peritonitis n = 3, colitis n = 1). Rats were used in most of the studies. Exhaled breath was mostly collected using invasive

**Data Availability Statement:** All relevant data are within the paper and its Supporting Information files.

**Funding:** Supported by ZonMW (114024163 to K. F.H.Hintzen) and the Dutch Digestive Foundation (MLDS career development grant CDG16-12 to T. Lubbers). The funders had no role in study design, data collection and analysis, decision to publish, or preparation of the manuscript.

**Competing interests:** The authors have declared that no competing interests exist.

procedures as tracheal cannulation or tracheostomy. Poor reporting on standardization, breath collection methods, analytical techniques, as well as heterogeneity of the studies, complicate comparison of the different studies.

## Conclusion

Poor reporting of essential methodological details impaired comprehensive summarizing the various studies on exhaled breath in gastrointestinal and hepatic diseases. Potential pitfalls in study design, and suggestions for improvement of study design are discussed which, when applied, lead to consistent and generalizable results and a reduction in the use of laboratory animals. Refining the methodological quality of animal studies has the potential to improve subsequent clinical trial design.

## Introduction

The discovery of a large number of volatile organic components (VOCs) in breath using gas-liquid partition chromatography (GC) by Linus Pauling marked an important step in the field of breath analysis [1]. VOCs are carbon-based organic molecules that can be detected in breath, sweat, blood, tissue samples, urine and feces [2, 3]. The secreted VOCs reflect the current metabolism of the sampled organism and alterations in this metabolism due to (patho) physiological processes result in marked changes in their composition. The specific characteristics of VOCs and the non-invasive character of the sampling process makes them an interesting target for the diagnosis of various diseases and monitoring of therapeutic interventions. In the past decades, there has been an increasing interest in the applicability of VOCs predominantly in breath as a rapid diagnostic tool in humans and animals [4].

Assessment of gastrointestinal and hepatic diseases currently often include invasive procedures such as endoscopy with the inevitable associated risks of complications [5]. As the gastrointestinal tract and the liver plays a major role in metabolism, pathological processes lead to metabolic changes and consequently affect the compositions of VOCs excreted in exhaled breath [6–8]. Therefore, breath analysis might provide a minimally invasive method for detection and monitoring of diseases. For example, the activity of inflammatory bowel disease (IBD) is correlated with breath alkanes, being a metabolic product of lipid peroxidation [9, 10]. For liver diseases, such as liver cirrhosis and hepatocellular carcinoma (HCC) an increased exhaled limonene is observed [11]. As a result, analysis of exhaled breath can provide insight in the different types and stages of gastrointestinal and liver diseases [12, 13]. However, despite these advantages and many years of scientific experience, VOCs are sparsely used in current clinical practice.

A major limitation of clinical breath studies is the heterogeneity in the results due to inappropriate study designs, inadequate phenotyping of patient groups, difficulties in standardization of sampling procedures and the lack of external validation [7, 12, 14–16]. Attempts to obtain standardized breath samples are often complicated by patient-related or environmental factors resulting in a large inter- and intra-individual variability. Various environmental factors can influence the composition of VOCs, including (polluted) air, diet, the use of consumer products like make-up or soaps, or smoking [3, 17]. Additionally, in the clinical setting anesthetics and disinfectants can also contribute to background-VOCs pollution in ambient air

[18, 19]. Overall, due to the heterogeneous results in human studies, it is difficult to relate specific VOCs to particular metabolic disorders or pathological processes.

In that respect, animal studies can play an important role in the preclinical research phase as they provide standardized disease models and confounding factors such as sex, comorbidities and diet can be kept to a minimum. Studies can be performed in a shorter period as target numbers of diseased subjects can be reached earlier. Standardized disease models and optimized sampling procedures allow for a better understanding of how pathological processes in a host lead to alterations in the composition of VOCs in exhaled breath. Therefore, animal studies can serve as proof-of-principle and provide insights that can be used to improve the design of clinical studies. However, collecting exhaled breath in animal studies can be challenging compared to human studies as animals can not be instructed. As a result, in the past different breath collection techniques have been described in literature, using animals that are either awake or anesthetized. Whole-body chambers have been used to collect air from the headspace of animals but this may be associated with contamination of VOCs from fur or animal excrement, potentially interfering with measurements of exhaled VOCs [20]. On the other hand, nose-only techniques try to prevent this contamination [21, 22]. Furthermore, anesthetics and tracheal cannulation or tracheostomies are used to collect exhaled breath during ventilation, but lead to the inability for long term follow-up [23–27].

The aim of this review is to create an overview of previously published animal studies on the methods for analysis of exhaled breath in gastrointestinal diseases, including liver pathology, and to assess the methodological quality of the conducted studies. A clear overview of the VOCs-related animal studies may potentially results in a better standardization of protocols for future research, reduction in the duplication of results, and reduction in the number of laboratory animals used. Moreover, standardization of protocols will enable a better comparison between studies, culminating in more targeted human research to evaluate the potential of exhaled air analysis in the clinical setting.

## Methods

This systematic review investigated animal studies into the analysis of exhaled breath for the diagnosis and screening of gastrointestinal and liver disease. The review methodology was specified in advanced and published in PROSPERO [CRD 42020208127].

### Search strategy and selection criteria

An electronic literature search was performed in Pubmed- and Embase library in December 2021 and was updated in March 2023. The search consisted out of 3 search components; 1) animal, 2) VOC and 3) gastrointestinal and liver disease. The SYRCLE PubMed [28] and Embase [29] search filters for animal studies were used to identify all experimental animal studies. All human benign and malignant gastrointestinal and liver diseases, breath collection, VOCs, electronic nose (e-nose), nano artificial nose (NA nose), breath biopsy and exhaled breath analysis were added as search terms and combined by using AND-OR combinations (S1 Checklist). Search results from both databases were combined and duplicates were removed. Furthermore, the reference list of the included studies were screened for additional relevant publications. Veterinary studies were excluded, as they do not concern human diseases. Human studies were also excluded, as the review focuses on VOCs in animal studies. Additionally, studies on sepsis, other than abdominal origin, and studies describing VOCs derived from other tissues like feces, blood, skin, urine and saliva were excluded. Studies in which VOCs were used to determine the liver function were excluded as these tests are a reflection of liver function, but do not lead to a single diagnosis. Studies were screened by two

independent reviewers (ME, CN) and disagreements were resolved by consensus after discussion with the third researcher (KH).

## Data extraction

The following data was collected and checked by two researchers (ME, KH): author(s), year of publication, number of animals, experimental groups, standardization (diet, fasting, day/night rhythm, environmental air), characteristics of the animals (species, sex, age), method of disease induction, reference test, validation, sampling and analytical techniques, sensitivity, specificity and identified VOCs.

## Risk of bias assessment

The SYRCLE's risk of bias tool, specific for animal studies, was used to assess the risk of bias in the selected studies. It contains 10 items related to selection bias, performance bias, detection bias, attrition bias, reporting bias and other biases (16). For each item, signaling questions are described to enhance transparency and applicability. These questions can be scored as 'yes', 'no' or 'unclear' resulting in an overall conclusion of the different types of bias. A '-' score indicates a high risk of bias; a '+' score indicates a low risk of bias; and a '?' score indicates an unknown risk of bias. The SYRCLE's risk of bias tool [30] was modified for the studies in which there was no control group or animals were used as their own control. Three reporting questions on randomization, blinding and power analysis were added. Experimental details on animals, materials and methods, are often poorly reported, resulting in scoring a high number of items as 'unclear risk of bias' [31]. Therefore, three additional items regarding reporting on randomization, blinding and power analysis, were added. For these three items, a 'yes' score indicates 'reported', and a 'no' score indicates 'unreported'. Two independent reviewers (ME, KH) assessed the risk of bias in the included paper and discrepancies were solved in a consensus meeting. Cohen's kappa (κ) was calculated for as level of agreement for the risk of bias. A separate kappa was calculated for the additional items regarding the reporting quality, as there were only two variables.

# Results

## Search

In September 2020, a total of 9039 studies were identified by the comprehensive search in Pubmed and Embase. After removing duplicates 7008 and full text analysis, a total of 10 studies were included in this review [32–41]. The updated search, in March 2023, resulted in the identification of 2009 extra studies. None of these additional studies were included (Fig 1).

## Study characteristics

The study characteristics containing information about study design, animal model characteristics and breath collection methods are summarized in Table 1. A wide variety of gastrointestinal and liver diseases is included in the studies. Four of the 10 studies involved liver diseases including, acute liver failure (n = 1) [41], non-alcoholic steatohepatitis (n = 1) [32] and hepatic ischemia (n = 2) [37, 40]. The studies regarding intestinal diseases focus on mesenteric ischemia (n = 2) [36, 39], sepsis and peritonitis (n = 3) [33–35] and colitis (n = 1) [38].

**Study design.**   Overall, the sample size ranged from 5 to 56 animals. Only 6 of the studies used a separate control group [32–35, 38, 39] whereas the other studies used animals as their own control collecting breath samples prior to disease induction [36, 37, 40, 41]. Standardization of diet, fasting status, day/night rhythm and environmental air varied between the

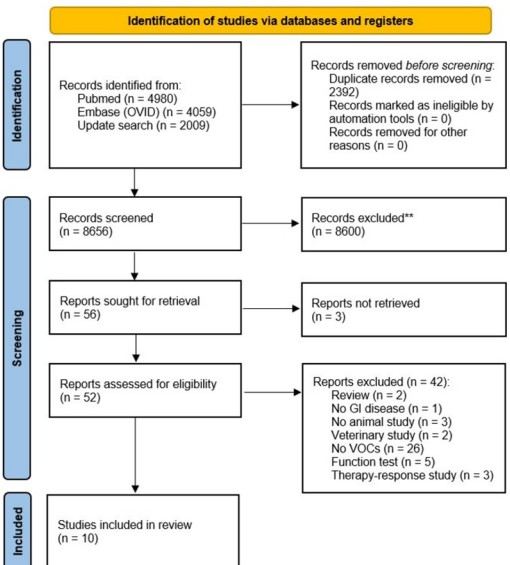

**Fig 1. PRISMA diagram showing the selection of the studies.**

included studies. The type of diet has been described in 6 of the 10 (60%) studies. For 1 of these studies, the diet was specifically adapted to a high-fat diet in order to induce non-alcoholic steatohepatitis [32]. In the other 5 studies, all animal had ad libitum access to a standard chow and water [33, 34, 37, 39, 41]. A period of fasting, prior to breath collection, was applied in 4 of the 10 (40%) studies [34, 37, 39, 40]. In all of these studies fasting consisted of 12 hours food restriction, but unlimited access to water. Four studies did not apply fasting prior to breath collection [32, 33, 35, 38] and in 2 studies no further information was provided [36, 41].

Standardization of the day/night rhythm is described in 4 of the 10 studies using 12h/12h light/dark cycles [32, 35, 37, 40]. In only 3 studies, measures were taken to limit possible influences of environmental air by either by collecting and analyzing the room air [33], collecting both room air and blanks of the sampling system [35] or using highly purified synthetic air for ventilation of the experimental animal [34].

**Animal model characteristics.** The majority of the studies (80%) included rats as experimental animals, using Wistar (n = 4) [32, 33, 36, 41] or Sprague-Dawley (n = 4) [34, 35, 38, 39] rats. One study on mesenteric ischemia used both rats and pigs in their experiments [39]. The remaining two studies used pigs [40] or rabbits [37] to study hepatic ischemia. Studies used male animals (n = 8) [32, 34, 35, 37–41] or no information regarding the sex of the animals was provided (n = 2) [33, 36]. Furthermore, the age of the animals was only mentioned in 3 studies and ranged from 8–20 weeks for rats [33, 38] and 3–3.5 months for pigs [40]. Since several different diseases are investigated in the included studies, the method of disease induction differs but relate to the specific disease of interest. All studies used a reference test to establish that the disease was properly induced.

**Breath collection methods.** 70% of the studies (n = 7) reported the use of anesthetics while collecting exhaled air via a tracheal cannula or tracheostomy [33–37, 39, 40]. Although the method of anesthesia differs, all studies describe the specific anesthetic agent that was used. Studies that collected exhaled breath in animals that are awake used either a nose-only sampling technique [32, 41] or a whole body technique [38, 39]. Nose-only techniques consisted of collecting exhaled breath directly from the muzzle. Animals where either placed in a tube with an opening for the muzzle, where air was passed and collected [41], or a tube was placed over

**Table 1. Study characteristics.**

| Study | Number of animals | Groups | Standardization | | | | Species | Sex | Age | Disease induction | Sample technique | | | Reference test | Analytical technique |
|---|---|---|---|---|---|---|---|---|---|---|---|---|---|---|---|
| | | | Diet | Fasting | Day/night rhythm | Air | | | | | Anesthetics | Tracheotomy / intubation | Nose-only vs whole body | | |
| *Acute liver failure (n = 1)* | | | | | | | | | | | | | | | |
| Wlodzimirow et al., 2014 | 14 | OC | SD[1] | NA | NA | NA | Wistar rat | M | NA | Portacaval shunt + ligation of hepatic artery and common bile duct | NA | NA | Nose-only | Yes (blood samples) | e-nose + GC-MS |
| Non-alcoholic steatohepatitis (n = 1) | | | | | | | | | | | | | | | |
| Aprea et al., 2012 | 16 | CG | NSD[2] | No | 12H | NA | Wistar rat | M | NA | High-fat diet | NA | NA | Nose-only | Yes (blood samples) | PTR-TOF-MS |
| *Hepatic ischemia (n = 2)* | | | | | | | | | | | | | | | |
| Liu et al., 2012 | 45 | OC | SD[3] | Yes* | 12H | NA | Rabbit | M | NA | Ligation of hepatic pedicle | Pentobarbital[I] | Tracheotomy and cannula | | Yes (blood samples) | SPME-GC-MS |
| Wang et al., 2012 | 20 | OC | NA | Yes* | 12H | NA | Swine | M | 3–3.5 months | Occlusion of portal inflow vessels (clamping) | Propofol or chloral hydrate[II] | Tracheal intubation | | Yes (blood samples & histopathology of liver tissue) | SPME-GC-MS |
| *Mesenteric ischemia (n = 2)* | | | | | | | | | | | | | | | |
| Jimenez et al., 2011 | 5 | OC | NA | NA | NA | NA | Wistar rat | NA | NA | Occlusion of superior mesenteric artery | Pentobarbital[III] | Tracheotomy and cannula | | Yes (visual examination of small intestine) | Gas chromatography |
| Szűcs et al., 2019 | 50 | CG | SD[4] | Yes* | NA | NA | Sprague-Dawly rat | M | NA | Tourniquet around superior mesenteric artery | Pentobarbital[IV] | Tracheotomy and cannula | Whole-body | Yes (tissue samples of ileum) | Photoacoustic spectroscopy |
| | 6 | | SD[4] | Yes* | NA | NA | Vietnamese minipig | NA | NA | Tourniquet around superior mesenteric artery | Propofol[V] | Tracheal intubation | | NA | Photoacoustic spectroscopy |
| *Sepsis and peritonitis (n = 3)* | | | | | | | | | | | | | | | |
| DeLano et al., 2017 | 24 | CG | SD[5] | No | NA | Room air samples as controls | Wistar rat | NA | 15–20 weeks | Cecal material injected in peritoneal cavity | Pentobarbital[VI] | Tracheotomy and cannula | | NA | Gas chromatography |
| Fink et al., 2015 | 40 | CG | SD[6] | Yes* | NA | Synthetic air | Sprague-Dawly rat | M | NA | Cecal ligation and incision (n = 10); Sham operation (n = 10); Endotoxemia by iv. injection of lipopolysaccharide (n = 10); Hemorrhagic shock (n = 10) | Pentobarbital[VII] | Tracheotomy and cannula | | Yes (blood samples) | MCC-IMS |
| Guamán et al., 2012 | 20 | CG | NA | No | 12H | Room air | Sprague-Dawly rat | M | NA | Injection with lipopolysaccharide from E. Coli | Xylazine & ketamine[IX] | Tracheotomy and cannula | | Yes (blood samples) | SPME-GC-MS + IMS |
| *Colitis (n = 1)* | | | | | | | | | | | | | | | |
| Ondrula et al., 1993 | 33 | CG | NA | No | NA | NA | Sprageue-Dawly rat | M | 8 weeks | Injection with trinitrobenzene-sulfonic acid and ethanol | NA | NA | Whole-body | Yes (macroscopy and histopathology of colon) | Gas chromatography |

***Abbreviations:*** NA = not available, OC = own control, CG = control group, SD = standard diet, NSD = no standard diet, 12H = 12-h light/dark cycle, M = male, iv. = intravenous, ip. = intraperitoneal, e-nose = electronic nose, GC-MS = gas chromatography mass spectrometry, PTR-TOF-MS = proton transfer reaction time of flight mass spectrometry, SPME-GC-MS = solid phase microextraction gas chromatography mass spectrometry, MCC-IMS = multicapillary column ion-mobility spectrometry, IMS = ion-mobility spectrometry.

[1] Standard rat chow and water ad libitum

[2] N = 4 HFD + W; n = 4 HFD + C; n = 4 SD + W; n = 4 SD + C

[3] Food and water were available ad libitum

[4] Standard laboratory chow with tap water ad libitum

[5] Regular diet (Harlan Teklad Rodent diet (W) 8604, 0.29% sodium by weight) without restriction and water ad libitum

[6] Standard pellet food and water ad libitum

* Fasting for 12 hours before the procedures with access to water

[I] 3% pentobarbital sodium solution 30 mg/kg iv.; rocuronium bromide 1 mg/kg iv.; fentany 10 ttg/kg iv.

[II] Premedication: ketamine 10 mg/kg im.; diazepam 0.2 mg/kg i.m.; atropine 0.05 mg/kg im.

Propofol group: induction with propofol 1.5 mg/kg iv.; maintenance with propofol 8–10 mg/kg/h, fentanyl 5 mg/kg & rocuronium 1 mg/kg

Chloral hydrate group: induction with chloral hydrate 0.5 g/kg iv.; maintenance with chloral hydrate 25–30 g/kg/h, fentanyl 5 mg/kg & rocuronium 1 mg/kg

[III] No additional information available

[IV] Sodium pentobarbital 50mg/kg ip.

[V] Induction with ketamine 20mg/kg im. & xylazine 2 mg/kg im.; maintenance with propofol 6mg/kg/h iv., midazolam 1.2mg/kg/hr iv. & fentanyl 0.02mg/kg/h iv.

[VI] Pentobarbital sodium 50 mg/kg

[VII] Pentobarbital 60mg/kg ip.

[IX] Xylazine 0.7 mL/kg ip. & ketamine 1ml/kg ip.

the muzzle while fixating the animal [32]. For the whole body technique animals were placed in a sealed glass cylinder (37) or a closed breath-collection chamber (36).

**Exhaled breath analysis.** Different analytical technique including gas chromatography (GC) alone [33, 36, 38] or combined with mass spectrometry (GC-MS) [37, 40, 41], solid proton transfer reaction time of flight mass spectrometry (PTR-TOF-MS) [32], photoacoustic spectroscopy [39], multicapillary column ion-mobility spectrometry (MCC-IMS) [34], ion-mobility spectrometry (IMS) [35], solid phase microextraction gas chromatography-mass spectrometry (SPME-GC-MS) [35] or electronic nose (e-nose) [41] technologies were used to analyze exhaled breath in the included studies. Two studies used 2 techniques simultaneously to further explore results and compare the techniques [35, 41]. The first study analyzed the breath samples with IMS and compared the results with SPME-GC-MS measurements as

**Table 2. Outcome measures.**

| Study | Sensitivity (%) | Specificity (%) | Identified VOCs |
|---|---|---|---|
| Acute liver failure (n = 1) | | | |
| Wlodzimirow et al., 2014 | e-nose Leave-one-out validation 89% | e-nose Leave-one-out validation 100% | 2-butanol; 2-butanone; 2-pentanone; 1-propanol; phenol; ethanol; dimethyl sulphide; hexane; pentane |
| Non-alcoholic steatohepatitis (n = 1) | | | |
| Aprea et al., 2012 | NA | NA | Acetonitrile; ammonia; dimethyl sulphide; dimethyl sulphone; methanol; phenol |
| Hepatic ischemia (n = 2) | | | |
| Liu et al., 2012 | NA | NA | Pentane |
| Wang et al., 2012 | NA | NA | Pentane |
| Mesenteric ischemia (n = 2) | | | |
| Jimenez et al., 2011 | NA | NA | Z,Z-farnesol; germacrene A; (Z,Z)-4,6,8-megastigmatriene; E,E-alpha-farnesene; delta-1-octene-3-ol; Z,E-farnesyl acetate; 2S-Z4Z7-13Ac; Z3-12OH; Z-tagetone; Z8-12Ac; 10Ald; geranial |
| Szűcs et al., 2019 | NA | NA | CH4 (methane) |
| Sepsis and peritonitis (n = 3) | | | |
| DeLano et al., 2017 | NA | NA | Acetone; amine; n-butyl acetate; 2-butanone; t-butanol; butyraldehyde; benzaldehyde; cadaverine; carbon disulfide; chlorobenzene; dimethyl sulfide; 1,4-diaminobutane; 1,5-diaminopentane; 1,2-dichloroethane; ethanol; indole; limonene; 6-methyl-5-hepten-2-one; 4-methylphenol; penta-noic acid; 1-propanol; putrescine; methylamine N-oxide, TMAO; o-xylene |
| Fink et al., 2015 | NA | NA | 1-Propanol, butanal, acetophenone, 1,2-butandiol, 3-pentanone, aceton, 2-hexanone |
| Guamán et al., 2012 | 98% (97.5–98.5%) | 85% (84.6–87.6%) | Cyclohexane, methyl; acetone; CO2; pentafluoropropionamide; dimethylether, Retention time (18.57) Mazas(42,48,56); o-Xylene; hexane, 2,3,4-trimethyl-; octane, 4-methyl-; decane; 2-Propanol, 1,3-dichloro-; toluene; acetic acid; propane, 2-ethoxy-2-methyl-; benzene; silanediol, dimethyl-; cyclotrisiloxane, hexamethyl-; cyclotetrasiloxane, octamethyl-; ketanone |
| Colitis (n = 1) | | | |
| Ondrula et al., 1993 | NA | NA | Pentane |

Abbreviations: NA = Not available

reference technique [35]. The other study, using an e-nose technique, used additional GC-MS measurements to further identify the compounds associated with the specific disease [41].

## Outcome measures

Outcome measures are summarized in Table 2. Diagnostic performance was analysed using 7 different methods. Three studies used a targeted approach based on one specific compound such as pentane [37, 38, 40] or methane [39], whereas others [32–36, 41] analyzed the complete breath profile identifying a wide variety of compounds. All three studies using pentane aim to diagnose either liver or intestinal ischemia, highlighting its possible relation to this specific disease process.

Only 2 studies reported the sensitivity and specificity of the performance of the diagnostic VOCs or e-nose readout. A sensitivity of 89% and specificity of 100% was found after leave-one-out validation for the study using an e-nose technique for acute liver failure [41]. No sensitivity and specificity for the GC-MS measurements were provided as these measurements were only used to identify compounds in this study. For sepsis, a sensitivity of 98% (97.5–98.5%) and specificity of 85% (84.6–87.6) was found using SPME-GC/MS and a sensitivity of 99.8 (99.7–99.9%) and specificity of 99.6 (99.5–99.7%) using IMS [35].

## Quality assessment of the studies

The results of the quality assessment are summarized in Table 3. There was a good level of agreement between the two reviewers on the risk of bias (kappa ($\kappa$) = 0.761 (p<0.001)) and on the reporting quality (kappa ($\kappa$) = 0.812 (p<0.001)). Generally, experimental details on the type of animals used, materials and methods are poorly described in the included studies. Hence, many items were scored as 'unclear risk of bias'. The additional reporting quality items on description of any randomization and blinding methods used, and the presence of a power

**Table 3. Risk of bias.**

| | Selection bias | | | Performance bias | | Detection bias | | Attrition bias | Reporting bias | Other | Reporting Quality | | |
|---|---|---|---|---|---|---|---|---|---|---|---|---|---|
| | Sequence generation | Baseline characteristics | Allocation concealment | Random housing | Blinding of caregivers and/or investigators | Random outcome assessment | Blinding of outcome assessor | Incomplete outcome data | Selective outcome reporting | Other sources of bias | Did the experiment randomise at any level? | Did the study blind at any level? | Did the study conduct a power analyses? |
| Wlodzimirow et al., 2014 | NA | NA | NA | NA | + | NA | ? | ? | ? | + | No | No | No |
| Aprea et al., 2012 | ? | ? | ? | + | ? | ? | ? | ? | ? | + | Yes | No | No |
| Liu et al., 2012 | NA | NA | NA | NA | ? | NA | ? | ? | ? | + | Yes | No | No |
| Wang et al., 2012 | NA | NA | NA | NA | ? | NA | ? | ? | ? | + | No | No | No |
| Jimenez et al., 2011 | NA | NA | NA | NA | ? | NA | - | ? | ? | ? | No | No | No |
| Szűcs et al., 2019 | + | ? | + | ? | - | + | - | ? | ? | + | Yes | No | No |
| DeLano et al., 2017 | ? | ? | ? | ? | ? | ? | ? | ? | ? | ? | No | No | No |
| Fink et al., 2015 | ? | ? | ? | ? | ? | + | ? | - | ? | + | Yes | Yes | Yes |
| Guamán et al., 2012 | ? | ? | ? | ? | ? | ? | ? | + | ? | + | No | No | No |
| Ondrula et al., 1993 | ? | + | ? | ? | ? | + | ? | ? | ? | ? | No | Yes | No |

NA = not applicable (own control group)

Red -: High risk of bias; Green +: Low risk of bias; Yellow ?: Unclear; NA: Not applicable

analysis/sample size calculation confirmed the poor reporting quality as also here the majority of studies did not describe any of these items. Half of the studies [33, 35, 36, 40, 41] scored 'no' on all three reporting quality items. Only 1 study [34] reported 'yes' on all three items. Of the remaining studies, 3 [32, 37, 39] reported randomization at any level, and only and 1 [38] on blinding at any level.

## Discussion

This is the first systematic review providing an overview on the analysis of exhaled air as a diagnostic tool for gastrointestinal and liver diseases in animal studies. This review focused on a number of possible confounders that can be standardized relatively easy in animal research, thereby facilitating future research using animal models. Unfortunately, our data demonstrates that the amount of animal studies on exhaled breath for the detection of gastrointestinal and liver diseases is limited and the experimental details are often poorly described. The large heterogeneity regarding standardization, breath collection methods, and analytical techniques complicates the comparison of the different studies. Standardization of study design, animal model characteristics and breath collection methods could help improve the methodological quality and comparability of the studies. As a result, suggestions for improvement of the experimental protocol and potential pitfalls are discussed which can ultimately lead to consistent results and a reduction in the use of laboratory animals.

As there is no standard breath collection method in the included studies, a distinction can be made between collecting breath from awake or anesthetized animals. This already results in a different experimental design as studies using anesthetized animals mainly result in terminal experiments and therefore impair the ability for long term follow-up or the need for more animals. This invasive technique might even result in the loss of animals during tracheotomy or tracheal cannulation [25]. Another disadvantage of this technique is a possible influence of the anesthetic agent, that is metabolized by the body [42], or the respirator itself on the collected VOCs [43]. Although pentobarbital appears to be the most commonly used anesthetic in the included studies, different doses were used and several additional medications, including analgesics or muscle relaxants, were administered. Whole-body or nose-only sampling techniques do not require anesthetics as they collect exhaled breath from animals that are awake. However, a whole-body technique results in contamination of VOCs originating from fur, feces or urine as the animal is placed in a respiratory chamber where the headspace of the animal is collected. This risk is limited if a nose-only technique, where the animal is fixated and breath is collected from only the muzzle, is used. However, handling and fixating the animal will inevitably lead to an increased stress response in the animals. Habituation and training of the animals is advisable when using nose-only techniques. Recently, we developed a device using the nose-only technique and performing a lung wash out with cleaned air, demonstrating the feasibility of non-invasive breath collection in murine models [44]. A major advantage of this technique is the ability for multiple breath collections from the same animal, eventually leading to a reduction of experimental animals needed.

One important factor, that is equally important in animal studies as in clinical studies, is the inhaled ambient air. The number of studies in which the influence of environmental air taken into account is currently very limited (n = 3) and the exact duration of the lung wash out period has not been defined. In animal experiments in general, multiple animals are housed together and exogenous VOCs from bedding material, food, feces, urine or other animals may contaminate exhaled breath. The amount of contamination may vary depending on the breath collection method used, as collecting exhaled breath via a respirator will be less sensitive to contamination of fur or feces compared to a whole body chamber, where the headspace of the

complete animal is collected. However, anesthetics or the respirator itself might also be a possible source of contamination [43]. Collecting and analyzing background samples as reference or performing a lung washout, in which clean or purified air is inhaled for a certain period, might help limiting these influences [45].

Differences in the composition of diet will influence the metabolic functions of the animal and therefore have an effect on the exhaled VOCs [46]. Not only different types and brands of diet, but also differences between batches of the same brand, due to growth conditions, may influence results [47]. Although 6 of the 10 studies described the type of diet as 'standard', further details on brands or specific composition were mostly not provided. Although it may not be feasible to use food from the same batch in different studies, reporting on brand names and food composition can already improve comparability. For single studies, it is strongly recommended to provide food from the same batch. In addition to the specific type of food, the fasting state of the animal may also affect its metabolic state [48]. This can be reduced by applying a fasting period prior to testing, which has already been described in half of the studies.

Next to diet, the circadian rhythm might have an influence on the activity and therefore the metabolism of the animals [49]. Animals that are sleeping or in a resting phase will be less active and will have less or no food intake which will inevitably affect the metabolic state and subsequently the composition of VOCs in exhaled breath. For example, isoprene is strongly correlated to exercise and shows a rapid increase or decrease in relation to movements, and an increase in acetone is related to a fasting state [50]. Most of the studies did not provide any details on measures taken to limit influences of the day/night rhythm. Studies that did describe the day/night rhythm all maintained a cycle of 12 hours of light and 12 hours dark [32, 35, 37, 40]. The influence of the circadian rhythm on exhaled VOCs varies during the day [51]. Collecting breath samples during the same period of the day might already help limit the influences of the circadian rhythm.

Additionally, the circadian rhythm is important factor for the regulation of adrenal glucocorticoid secretion involved in the homeostatic response to stress. This response differs between sexes and may be subject to fluctuations in gonadal hormones or during aging [51, 52]. Therefore, it is recommended to take age and gender into account when performing exhaled breath research and to standardize these factors if possible.

Identification of disease specific VOCs, independent of host- or environmental factors, remains difficult as identified VOCs in vivo may not always emerge in clinical studies or clinical studies may reveal different VOCs. The included studies have identified several VOCs of disease processes, some of which are described repeatedly. Pentane is described in multiple included animal studies in liver disease [1–3] and is known to be the result of cytochrome P450 activity and is metabolized in the liver indicating its relation to liver (dys)function [4]. However, other studies show its relation to lipid peroxidation and inflammatory bowel disease as well [5, 6], which has been described in the included study on colitis [7]. An increase or decrease in the presence of pentane may thus be affected by both alterations in its production or metabolic loss. Another compound that is frequently described is acetone, which is an endogenous ketone related to fatty acid oxidation. Alterations in its presence has been described as the result of inflammatory of infectious states [8] and also in clinical studies on gastro-intestinal malignancies [9]. In this review, acetone has been identified in all three studies on sepsis and peritonitis which could be explained by its known relation to inflammatory states. However, various (patho-)physiological processes involved in the increase or decrease of VOCs in combination with host- or environmental factors makes it difficult to identify specific compounds.

Human studies clearly point out the need for larger studies, standardization, validation, and a better mechanistic understanding of diseases [7, 8, 12, 53]. Animal studies might serve as

proof-of-principle or contribute to a first selection of disease specific VOCs before setting up large clinical studies. However, similar to human studies, animal studies should meet certain standards in order to be of sufficient methodological quality. Already existing guidelines, such as the ARRIVE guidelines [54] or the SYRCLE's risk of bias tool [30], provide general recommendations for reporting and or the conduct of animal studies. These tools can be used in designing new studies in order to improve methodological quality, experimental design, reliability and comparability of the studies. Standardization is of great importance in breath research. Therefore, the current study provides additional guidelines specifically for animal studies using exhaled breath analysis (Fig 2) to improve comparability, avoid duplication of research and reduce discomfort of animals in future studies. This guideline consists of three important domains, namely the animal model (A), the breath collection (B) and study design (C). When choosing an animal model (A), the researcher should A1) consider the type of species and strain that is most suitable for the disease model by reviewing research on the specific disease; A2) standardize for sex and age/weight if applicable and; A3) use comparable disease induction methods. For the exhaled breath collection and analysis (B) both B1) sampling techniques and B2) analytical techniques should be standardized. As for study design (C); C1) type of diet and fasting state; C2) day/night rhythm; and C3) environmental air should be standardized and background air samples should be collected as reference; C4) air samples should be randomly collected and analyzed and C5) blinding of researchers and outcome assessors should be applied as much as possible within the possibilities of the experiments; C6 study design and results should be reported in a clear and transparent manner for reproducibility as already described in the ARRIVE guidelines [54].

Minimally invasive methods, such as nose-only techniques, are preferred if the specific disease model allows for it. Any medication, both anesthetics and other medication, provided to the animals should be described. It is of great importance that reporting and standardization of the possible confounders described above will be improved in future research, as these factors might have a major influence on VOCs in exhaled breath and the comparability of the studies. Animal studies are an important factor in the initial understanding of disease mechanisms, even though they do not provide a direct translation to clinical use. Improvements will

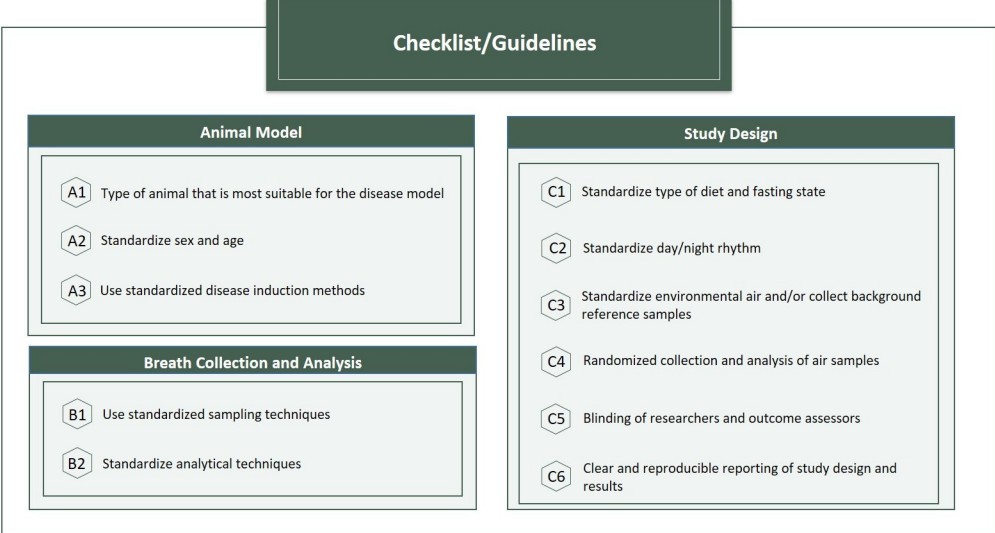

**Fig 2. Proposed guidelines for breath research in animal research.**

have to be made in standardization and reporting of methods in order to increase comparability and improve future translation to clinical research.

## Supporting information

**S1 Checklist. PRISMA checklist.**
(DOCX)

**S1 File. Search strategy.**
(DOCX)

## Author Contributions

**Conceptualization:** Kim F.H. Hintzen, Myrthe M.M. Eussen, Frederik-Jan van Schooten, Carlijn R. Hooijmans, Tim Lubbers.

**Data curation:** Kim F.H. Hintzen, Myrthe M.M. Eussen, Céline Neutel, Carlijn R. Hooijmans.

**Formal analysis:** Kim F.H. Hintzen, Myrthe M.M. Eussen.

**Funding acquisition:** Kim F.H. Hintzen, Myrthe M.M. Eussen, Tim Lubbers.

**Investigation:** Kim F.H. Hintzen, Myrthe M.M. Eussen, Céline Neutel.

**Methodology:** Kim F.H. Hintzen, Myrthe M.M. Eussen, Carlijn R. Hooijmans.

**Project administration:** Tim Lubbers.

**Resources:** Frederik-Jan van Schooten.

**Supervision:** Nicole D. Bouvy, Frederik-Jan van Schooten, Carlijn R. Hooijmans, Tim Lubbers.

**Visualization:** Kim F.H. Hintzen.

**Writing – original draft:** Kim F.H. Hintzen, Myrthe M.M. Eussen, Céline Neutel.

**Writing – review & editing:** Kim F.H. Hintzen, Nicole D. Bouvy, Frederik-Jan van Schooten, Carlijn R. Hooijmans, Tim Lubbers.

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
