## [Decision Letter · Decision Letter 0]

6 Jun 2023

PONE-D-23-11029A systematic review on the detection of volatile organic compounds in exhaled breath in experimental animals in the context of gastrointestinal diseasesPLOS ONE

Dear Dr. Hintzen,

Thank you for submitting your manuscript to PLOS ONE. After careful consideration, we feel that it has merit but does not fully meet PLOS ONE’s publication criteria as it currently stands. Therefore, we invite you to submit a revised version of the manuscript that addresses the points raised during the review process.

 Each reviewer has made a number of minor points, all of which should be addressed in the manuscript or in your response to reviewers' comments. 

We look forward to receiving your revised manuscript.

Kind regards,

Christopher Walton, Ph.D.

Academic Editor

PLOS ONE

“Supported by ZonMW (114024163 to K.F.H.Hintzen) and the Dutch Digestive

Foundation (MLDS career development grant CDG16-12 to T. Lubbers)”

Please respond by return e-mail so that we can amend your financial disclosure and competing interests on your behalf.

Reviewers' comments:

Reviewer's Responses to Questions

**Comments to the Author**

1. Is the manuscript technically sound, and do the data support the conclusions?

Reviewer #1: Yes

Reviewer #2: Yes

2. Has the statistical analysis been performed appropriately and rigorously? 

Reviewer #1: Yes

Reviewer #2: N/A

3. Have the authors made all data underlying the findings in their manuscript fully available?

Reviewer #1: Yes

Reviewer #2: Yes

4. Is the manuscript presented in an intelligible fashion and written in standard English?

Reviewer #1: Yes

Reviewer #2: Yes

5. Review Comments to the Author

Reviewer #1: This rigorous review will benefit readers in the breath analysis field. The manuscript is well written and properly following the PRISMA systematic search. It was very interesting and enjoyable to read.

Some suggestions to improve the content:

1) The answers of both reviewers (ME, KH) should be included in the results of the quality assessment. In addition, Cohen’s kappa (k) should be determined to evaluate the level of agreement in the responses of reviewers.

2) I would recommend including a specific section about the reported VOCs in the selected studies, where possible sources of theses VOCs are discussed and related with finding in human exhaled breath (DOI: 10.1111/apt.13050; DOI:10.1016/j.ebiom.2020.102725; DOI: 10.1038/s41390-020-01116-8).

Reviewer #2: The systematic review submitted by Hintzen summarize and critically evaluate the studies of VOC analysis for the detection of gastrointestinal and hepatic diseases. I believe that the review will be interesting to the audience of PLOS ONE.

I have several remarks concerning the review:

1. The review comprises studies on of gastrointestinal and hepatic diseases, and not only on of gastrointestinal diseases as claimed in the title, abstract and text. This should be corrected.

2. The authors emphasise on the need of standardization of breath research methodologies, e.g. sampling and analytical techniques, in order to increase their comparability. However, heterogenic methodologies can be compared on the basis of sensitivity, selectivity and accuracy they offer for disease detection. Moreover, heterogenic approaches can help to identify robust disease biomarkers, independent of the sampling and analytical techniques. Therefore, the authors should clarify what standardization they have in mind.

3. I strongly agree that there is need for standardization of the demands for study details description in publication. Thus, I propose to modify the title of figure 2 to “Proposed guidelines/checklist for breath research data report in animal studies” and modify the figure appropriately.

4. In the fig. 2 domain B concerns the analytical methodology as a whole, not just sampling (Breath collection), so I think the title of the domain should be modified.

Minor remarks:

1. PTR-tof-MS should be changed to PTR-TOF-MS

2. In table 1 Seks is obviously Sex

3. In table 1 “Breath collection methods” should be changed to “Analytical Methodology”

4. In Table 1 when Gas Chromatography indicated as analytical technique the detector used should be also indicated, e.g. GC-FID

6. PLOS authors have the option to publish the peer review history of their article (what does this mean?). If published, this will include your full peer review and any attached files.

Reviewer #1: No

Reviewer #2: No

---

## [Author Response · Author response to Decision Letter 0]

29 Aug 2023

Dear dr. Walton,

Thank you for forwarding the response of the referees on our manuscript PONE-D-23-11029 entitled “A systematic review on the detection of volatile organic compounds in exhaled breath in experimental animals in the context of gastrointestinal diseases“. 

We would like to thank the reviewers for their encouraging words. Their suggestions for improvements are a valuable addition and were used for minor revisions of the manuscript. 

Please find our response to the comments of the reviewers below. 

Reviewer #1: 

This rigorous review will benefit readers in the breath analysis field. The manuscript is well written and properly following the PRISMA systematic search. It was very interesting and enjoyable to read.

Some suggestions to improve the content:

1) The answers of both reviewers (ME, KH) should be included in the results of the quality assessment. In addition, Cohen’s kappa (k) should be determined to evaluate the level of agreement in the responses of reviewers.

Thank you for this comment and the suggestion to determine the Cohen’s kappa (k) to the review as an objective level of agreement between the reviewers. We have calculated this and made the adjustments in the manuscript in the methods on page 6, line 15 and in the results on page 16, line 2. 

Methods:

“Cohen’s kappa (k) was calculated as level of agreement for the risk of bias. A separate kappa was calculated for the additional items regarding the reporting quality, as there were only two variables.”

Results: 

“There was a good level of agreement between the two reviewers on the risk of bias (kappa (κ) = 0.761 (p<0.001)) and on the reporting quality (kappa (κ) = 0.812 (p<0.001)).”

2) I would recommend including a specific section about the reported VOCs in the selected studies, where possible sources of theses VOCs are discussed and related with finding in human exhaled breath (DOI: 10.1111/apt.13050; DOI:10.1016/j.ebiom.2020.102725; DOI: 10.1038/s41390-020-01116-8).

We agree that it could be a valuable addition to discuss possible sources of identified VOCs, especially related with findings in human exhaled breath. However, the aim of this review is not to provide a selection of disease specific VOCs but to provide an overview of previously published animal studies, using exhaled breath in gastro-intestinal and liver diseases, in order to improve the design of future research. As mentioned in the introduction, page 4 line 1, it remains difficult to relate specific VOCs to particular metabolic disorders or pathological processes. To emphasize difficulties in the translation to clinical studies, and the sometimes insufficient fundamental understanding of how pathological processes in a host lead to alterations of VOCs in exhaled breath, we added the following part in the discussion on page 21, line 12 onwards:

“Identification of disease specific VOCs, independent of host- or environmental factors, remains difficult as identified VOCs in vivo may not always emerge in clinical studies or clinical studies may reveal different VOCs. The included studies have identified several VOCs of disease processes, some of which are described repeatedly. Pentane is described in multiple included animal studies in liver disease [1-3] and is known to be the result of cytochrome P450 activity and is metabolized in the liver indicating its relation to liver (dys)function [4]. However, other studies show its relation to lipid peroxidation and inflammatory bowel disease as well [5, 6], which has been described in the included study on colitis [7]. An increase or decrease in the presence of pentane may thus be affected by both alterations in its production or metabolic loss. Another compound that is frequently described is acetone, which is an endogenous ketone related to fatty acid oxidation. Alterations in its presence has been described as the result of inflammatory of infectious states [8] and also in clinical studies on gastro-intestinal malignancies [9]. In this review, acetone has been identified in all three studies on sepsis and peritonitis which could be explained by its known relation to inflammatory states. However, various (patho-)physiological processes involved in the increase or decrease of VOCs in combination with host- or environmental factors makes it difficult to identify specific compounds.”

Reviewer #2: 

The systematic review submitted by Hintzen summarize and critically evaluate the studies of VOC analysis for the detection of gastrointestinal and hepatic diseases. I believe that the review will be interesting to the audience of PLOS ONE.

I have several remarks concerning the review:

1. The review comprises studies on of gastrointestinal and hepatic diseases, and not only on of gastrointestinal diseases as claimed in the title, abstract and text. This should be corrected.

It is indeed that, strictly speaking, liver diseases are not the same as gastrointestinal diseases. We have therefore implemented the suggested adjustment in title, abstract and text.

2. The authors emphasise on the need of standardization of breath research methodologies, e.g. sampling and analytical techniques, in order to increase their comparability. However, heterogenic methodologies can be compared on the basis of sensitivity, selectivity and accuracy they offer for disease detection. Moreover, heterogenic approaches can help to identify robust disease biomarkers, independent of the sampling and analytical techniques. Therefore, the authors should clarify what standardization they have in mind. 

Thank you for this critical comment. We do agree that using different analytical techniques and study designs contributes to the discovery of the more robust biomarkers. However, as demonstrated by the vast amount of in vivo and clinical publications on the detection of VOCs in exhaled air and limited clinical use of VOC analysis, too much heterogeneity between studies negatively influences comparability between studies and impairs identification of specific disease related VOCs. 

There is a wide range of factors that can influence the composition of VOCs in exhaled breath that, if not standardized, may lead to results in which it is unclear whether an identified VOC is related to the specific disease, stage of the disease, specific pathogen or to environmental or experimental factors. 

For example, in vitro research already showed that specific VOCs produced by bacteria can be subject to differences in growth conditions, such as type of culture media that is used [10] or to the dynamic growth process of bacteria [11].

We believe that animal studies are important precisely because of the possibilities for standardization and thus making results comparable. These standardized approaches can be hypothesis generating and will facilitate the design and analysis of clinical research. 

The guidelines described in figure 2 and elaborated on in the discussion from page 21, line 31 onwards should be used in the design of new research projects. The use of these guidelines and adequate reporting of the study design and results will ensure better reproducibility of the studies. As a result, future researchers will have the opportunity to repeat these studies or use these research models to examine different aspects of the specific disease. 

As for data report, it may be a good addition to report sensitivity, selectivity and accuracy to compare heterogeneous methodologies. However, these values are usually concerning a combination of VOCs rather than separate VOCs. In this review, only 2 out of the 10 studies provided a sensitivity, specificity or accuracy, as described in Table 2. One of these studies used an e-nose, which does not identify specific VOCs, and the other study used a combined set of VOCs. Using different analytical techniques with different detection ranges may result in overlooking certain VOCs or identifying additional VOCs compared to other techniques and therefore less comparable results. 

3. I strongly agree that there is need for standardization of the demands for study details description in publication. Thus, I propose to modify the title of figure 2 to “Proposed guidelines/checklist for breath research data report in animal studies” and modify the figure appropriately.

Thank you for your proposal to extend the applicability of the guidelines in Figure 2 to not only the design of new studies but also to improve reporting of these studies. However, it should be noted that general reporting guidelines for animal studies already exist, as described in the discussion on page 21, from line 32 onwards. Therefore, we have not included any new guidelines/checklist on reporting of animal studies. 

However, we have modified Figure 2 in which we now recommend to use of existing general guidelines for reporting animal studies “C6; clear and reproducible reporting of study design and results”. This way awareness of reporting guidelines can be increased. This may lead to more appropriate use of the existing guidelines and subsequently improving the reporting quality of animal studies, including breath research. 

We have added this in the manuscript text on page 22, line 16.

“C6 study design and results should be reported in a clear and transparent manner for reproducibility as already described in the ARRIVE guidelines [12].

4. In the fig. 2 domain B concerns the analytical methodology as a whole, not just sampling (Breath collection), so I think the title of the domain should be modified.

We agree that this domain concerns both the breath sampling technique and the analytical technology. Therefore, we adjusted the title of the domain to “Breath Collection and Analysis”. 

Additionally, we have changed “For the exhaled breath analysis” to “For the exhaled breath collection and analysis” in the discussion on page 22, line 11.

Minor remarks:

1. PTR-tof-MS should be changed to PTR-TOF-MS -> Adjusted in Table 1 and in the results on page 13, line 13

2. In table 1 Seks is obviously Sex -> Adjusted in Table 1

3. In table 1 “Breath collection methods” should be changed to “Analytical Methodology” -> Changes “Breath collection methods” tot “Breath collection and analytical methods” 

4. In Table 1 when Gas Chromatography indicated as analytical technique the detector used should be also indicated, e.g. GC-FID -> We have adjusted this in Table 1 and indicated the detector used for Jimenez et al. (2011), DeLano et at. (2017) and Ondrula et al. (1993). In the abbreviations we added the following text: “GC-SAW = gas chromatography surface acoustic wave detection, GC-FID = gas chromatography flame ionization detector”

References

1. Liu S, Shi J, Wang C, Li P, Gong Y, He Y, et al. Measurement of pentane in expiratory gas during rabbit hepatic ischemia/ reperfusion by solid-phase microextraction and gas chromatography–mass spectrometry (SPME GC/MS). Journal of breath research. 2012;6(2):026003.

2. Wang C, Shi J, Sun B, Liu D, Li P, Gong Y, et al. Breath pentane as a potential biomarker for survival in hepatic ischemia and reperfusion injury--a pilot study. PLoS One. 2012;7(9):e44940.

3. Wlodzimirow KA, Abu-Hanna A, Schultz MJ, Maas MA, Bos LD, Sterk PJ, et al. Exhaled breath analysis with electronic nose technology for detection of acute liver failure in rats. Biosens Bioelectron. 2014;53:129-34.

4. Stavropoulos G, van Munster K, Ferrandino G, Sauca M, Ponsioen C, van Schooten FJ, et al. Liver Impairment-The Potential Application of Volatile Organic Compounds in Hepatology. Metabolites. 2021;11(9).

5. Dryahina K, Španěl P, Pospíšilová V, Sovová K, Hrdlička L, Machková N, et al. Quantification of pentane in exhaled breath, a potential biomarker of bowel disease, using selected ion flow tube mass spectrometry. Rapid Commun Mass Spectrom. 2013;27(17):1983-92.

6. Kurada S, Alkhouri N, Fiocchi C, Dweik R, Rieder F. Review article: breath analysis in inflammatory bowel diseases. Aliment Pharmacol Ther. 2015;41(4):329-41.

7. Ondrula D, Nelson RL, Andrianopoulos G, Schwartz D, Abcarian H, Birnbaum A, et al. Quantitative determination of pentane in exhaled air correlates with colonic inflammation in the rat colitis model. Dis Colon Rectum. 1993;36(5):457-62.

8. Schnabel R, Fijten R, Smolinska A, Dallinga J, Boumans ML, Stobberingh E, et al. Analysis of volatile organic compounds in exhaled breath to diagnose ventilator-associated pneumonia. Sci Rep. 2015;5:17179.

9. Hintzen KFH, Grote J, Wintjens A, Lubbers T, Eussen MMM, van Schooten FJ, et al. Breath analysis for the detection of digestive tract malignancies: systematic review. BJS Open. 2021;5(2).

10. Boots AW, Smolinska A, van Berkel JJ, Fijten RR, Stobberingh EE, Boumans ML, et al. Identification of microorganisms based on headspace analysis of volatile organic compounds by gas chromatography-mass spectrometry. Journal of breath research. 2014;8(2):027106.

11. Filipiak W, Sponring A, Baur MM, Filipiak A, Ager C, Wiesenhofer H, et al. Molecular analysis of volatile metabolites released specifically by Staphylococcus aureus and Pseudomonas aeruginosa. BMC Microbiol. 2012;12:113.

12. Percie du Sert N, Hurst V, Ahluwalia A, Alam S, Avey MT, Baker M, et al. The ARRIVE guidelines 2.0: Updated guidelines for reporting animal research. Br J Pharmacol. 2020;177(16):3617-24.

---

## [Editor Report · Decision Letter 1]

4 Sep 2023

A systematic review on the detection of volatile organic compounds in exhaled breath in experimental animals in the context of gastrointestinal diseases

PONE-D-23-11029R1

Dear Dr. Hintzen,

We’re pleased to inform you that your manuscript has been judged scientifically suitable for publication and will be formally accepted for publication once it meets all outstanding technical requirements.

Kind regards,

Christopher Walton, Ph.D.

Academic Editor

PLOS ONE
---

## [Editor Report · Acceptance letter]

11 Sep 2023

PONE-D-23-11029R1 

A systematic review on the detection of volatile organic compounds in exhaled breath in experimental animals in the context of gastrointestinal and hepatic diseases 

Dear Dr. Hintzen:

I'm pleased to inform you that your manuscript has been deemed suitable for publication in PLOS ONE. Congratulations! Your manuscript is now with our production department. 

Kind regards, 

on behalf of

Dr. Christopher Walton 

Academic Editor

PLOS ONE